# Trastuzumab Conjugated Superparamagnetic Iron Oxide Nanoparticles Labeled with ^225^Ac as a Perspective Tool for Combined α-Radioimmunotherapy and Magnetic Hyperthermia of HER2-Positive Breast Cancer

**DOI:** 10.3390/molecules25051025

**Published:** 2020-02-25

**Authors:** Edyta Cędrowska, Marek Pruszyński, Weronika Gawęda, Michał Żuk, Paweł Krysiński, Frank Bruchertseifer, Alfred Morgenstern, Maria-Argyro Karageorgou, Penelope Bouziotis, Aleksander Bilewicz

**Affiliations:** 1Institute of Nuclear Chemistry and Technology, Dorodna 16, 03-195 Warsaw, Poland; e.leszczuk@ichtj.waw.pl (E.C.); w.maliszewska@ichtj.waw.pl (W.G.); 2Faculty of Chemistry, Biological and Chemical Research Centre, University of Warsaw, Żwirki i Wigury 101, 02-089 Warsaw, Poland; 3Faculty of Chemistry, University of Warsaw, Pasteura 1, 02-093 Warsaw, Poland; mt_zuk@chem.uw.edu.pl (M.Ż.); pakrys@chem.uw.edu.pl (P.K.); 4Department for Nuclear Safety and Security, Joint Research Centre, European Commission, 76125 Karlsruhe, Germany; Frank.BRUCHERTSEIFER@ec.europa.eu (F.B.); alfred.morgenstern@ec.europa.eu (A.M.); 5Institute of Nuclear & Radiological Sciences & Technology, Energy & Safety, N.C.S.R. ‘Demokritos’, Aghia Paraskevi, 15341 Athens, Greece; kmargo@phys.uoa.gr (M.-A.K.); bouzioti@rrp.demokritos.gr (P.B.); 6Department of Physics, National and Kapodistrian University of Athens, Zografou Panepistimioupolis, 15784 Athens, Greece

**Keywords:** radionuclide therapy, hyperthermia, SPION, trastuzumab

## Abstract

It has been proven and confirmed in numerous repeated tests, that the use of a combination of several therapeutic methods gives much better treatment results than in the case of separate therapies. Particularly promising is the combination of ionizing radiation and magnetic hyperthermia in one drug. To achieve this objective, magnetite nanoparticles have been modified in their core with α emitter ^225^Ac, in an amount affecting only slightly their magnetic properties. By 3-phosphonopropionic acid (CEPA) linker nanoparticles were conjugated covalently with trastuzumab (Herceptin^®^), a monoclonal antibody that recognizes ovarian and breast cancer cells overexpressing the HER2 receptors. The synthesized bioconjugates were characterized by transmission electron microscopy (TEM), Dynamic Light Scattering (DLS) measurement, thermogravimetric analysis (TGA) and application of ^131^I-labeled trastuzumab for quantification of the bound biomolecule. The obtained results show that one ^225^Ac@Fe_3_O_4_-CEPA-trastuzumab bioconjugate contains an average of 8–11 molecules of trastuzumab. The labeled nanoparticles almost quantitatively retain ^225^Ac (>98%) in phosphate-buffered saline (PBS) and physiological salt, and more than 90% of ^221^Fr and ^213^Bi over 10 days. In human serum after 10 days, the fraction of ^225^Ac released from ^225^Ac@Fe_3_O_4_ was still less than 2%, but the retention of ^221^Fr and ^213^Bi decreased to 70%. The synthesized ^225^Ac@Fe_3_O_4_-CEPA-trastuzumab bioconjugates have shown a high cytotoxic effect toward SKOV-3 ovarian cancer cells expressing HER2 receptor in-vitro. The in-vivo studies indicate that this bioconjugate exhibits properties suitable for the treatment of cancer cells by intratumoral or post-resection injection. The intravenous injection of the ^225^Ac@Fe_3_O_4_-CEPA-trastuzumab radiobioconjugate is excluded due to its high accumulation in the liver, lungs and spleen. Additionally, the high value of a specific absorption rate (SAR) allows its use in a new very perspective combination of α radionuclide therapy with magnetic hyperthermia.

## 1. Introduction

Nanotechnology has great potential in the development of medical diagnosis and therapy by designing and synthesizing a wide range of nanomaterials for applications in targeted drug delivery, accurate diagnosis, and treatment of diseases such as cancers [1]. Until now, various of nanoparticles (NPs) have been utilized as radioisotope carriers in radionuclide therapy, among them organic NPs such as polymeric matrixes [2], liposomes [3] and inorganic NPs such as gold [4], metal oxides [5] and insoluble salts of metals [6]. The advantage of NPs is their ability to “carry” many radioactive atoms within a single carrier. Additionally, nanoparticles are ideal platforms to incorporate multiple functions and enable multimodality in diagnosis and therapy. In some cases, such as ^223^Ra radiobioconjugates, nanotechnology is the only method for the stable coupling of a radionuclide to a biomolecule [7]. NPs can deliver radionuclides to the cancer tissue using passive or active targeting strategy. The passive accumulation of NPs in the tumor takes place across enlarged gap junctions in tumor endothelial cells [8]. This type of targeting, which enables macromolecules to selectively accumulate in the tumor tissue, is called Enhanced Permeation and Retention (EPR). Unfortunately, local drug deposition is unfeasible for larger tumors with poor vascularization and for floating cancers like lymphoma and leukemia. Target specificity is then achieved through modification of NPs by conjugation with tumor-specific biomolecules such as monoclonal antibodies (mAbs), aptamers, peptides, or various receptor-specific compounds [9]. Thus, the uptake of bioconjugated nanoparticles in cells with overexpressed receptors greatly increases. Unfortunately, the limitation of intravenously administrated NPs is their sequestration by the liver, lungs, and spleen, causing rapid elimination from the blood, radiation exposure of these organs and low tumor uptake [10].

Recently Reilly et al. [11,12] proposed a novel targeted nanomedicine brachytherapy approach for the treatment of locally-advanced breast cancer, which uses local intratumoral injection of 30-nm-diameter gold nanoparticles (AuNPs) modified with polyethylene glycol (PEG) chains derivatized with the 1,4,7,10-tetraazacyclododecane-1,4,7,10-tetraacetic acid (DOTA) chelator that complexes the therapeutic radionuclide ^177^Lu. Then, they linked the DOTA-functionalized NPs to panitumumab which allows binding of the obtained radionanoparticles selectively with Epidermal Growth Factor Receptor (EGFR) or linked it to trastuzumab that binds to the Human Epidermal Growth Factor Receptor type 2 (HER2), both receptors overexpressed on cancerous cells. Biodistribution studies in xenografted-mice showed that AuNPs-trastuzumab delivered intratumorally was retained ~30% ID/g) with minimal uptake by the liver and spleen. They also found that targeting HER2 with trastuzumab facilitated binding, while internalization of trastuzumab functionalized AuNPs was much higher compared to non-functionalized ^177^Lu@AuNPs [13]. (Nonobese diabetic) NOD/(severe combined immunodeficient) SCID mice bearing human breast cancer xenografts overexpressing HER2 demonstrated that tumor growth inhibition was significantly higher when compared to mice injected with ^177^Lu@AuNPs or untreated mice, over a 16-day period.

In our current work to increase the efficiency and selectivity of therapy, we proposed to apply a similar approach, but instead of the β^−^ emitter ^177^Lu, we used the much more radiotoxic alpha-emitter ^225^Ac. The range of α-particles in tissues corresponds to only 5–10 cell diameters, which limits the deposition of radiation to the targeted cell and closely neighboring cells. Compared with β^−^ particles, α particles provide a very high relative biological effectiveness, killing more cells with less radioactivity. The high linear energy transfer of α particles induces significantly more DNA double-strand breaks than β^−^ particles [14]. The biological effectiveness of α particles is also independent of tissue oxygenation, which is advantageous compared to β^−^ emitters for radioimmunotherapy, which depends on the formation of oxygen free radicals [15].

Various methods of attaching ^225^Ac to biomolecules have been proposed. In addition to cyclic and linear ligands like DOTA, HEHA or DTPA, they were various types of nanoparticles like gadolinium vanadate nanocrystals [16], polymersomes [17], indium and lanthanum phosphates [18,19]. In our work as a carrier for ^225^Ac, we proposed to use Superparamagnetic Iron Oxide-based Nanoparticles (SPIONs) where actinium similarly to lanthanides should be incorporated into their magnetic core [20]. SPIONs have attracted wide attention due to their unique properties originating from intrinsic magnetic properties (due to ferromagnetic iron), high surface area-to-volume ratio, and the ability to chemically modify their core composition and their surfaces. In previous works, SPIONs have been assembled with a series of radionuclides including ^18^F [21], ^59^Fe [22], ^64^Cu [23,24], ^89^Zr [25], ^99m^Tc [26], ^111^In [27] and ^223^Ra [28]. There are two strategies for the synthesis of radioactive SPIONs: incorporation into the structure or surface functionalization (including the construction of shell layers). The attractiveness of magnetic nanoparticles is that they exhibit high magnetization in an external magnetic field but none when the magnetic field is removed. The superparamagnetic behavior makes iron oxide-based nanoparticles ideal candidates as carrier platforms in targeted drug delivery, where the drug/carrier conjugate can be guided to the specific site with the aid of an external, constant magnetic field. In the case of targeted nanoparticle brachytherapy, an external constant magnetic field will allow retention of the radiobioconjugate in the region of the neoplastic lesion.

Futhermore, ^225^Ac-doped SPIONs are promising candidates for the multimodal effect of targeted drug delivery, combined localized magnetic hyperthermia and radionuclide therapy [29,30]. SPIONs localized in the tumor heat up under the effect of an alternating magnetic field (AMF), stimulating cancer tissue destruction [31]. Heating of nanoparticles up to 41–47 °C for dozens of minutes by hyperthermia triggers apoptosis [32,33], destabilizes cells and offsets homeostasis, leading to higher susceptibility of cancer cells to chemotherapy [34]. Hyperthermia can potentiate the effect of radiation and has been shown to improve local treatment in patients with advanced cancers like breast or head and neck [35] spread cancers. The synergetic effect of external ionizing radiation and hyperthermia in killing cancer cells is well-known since the discovery of cancer cells resistant to radiation but sensitive to hyperthermic conditions. Combining both therapeutic concepts, one expects an increased efficiency of the radiation treatment by prior application of hyperthermia which could possibly decrease the delivered radiation dose. It is thought that tumor tissue is more hypoxic, more acidic, and nutrient-deficient compared to normal tissues [36]. Because radiation resistance is often observed in the same tumor regions, the temperature rise increases the effectiveness of radiotherapy and allows the reduction of the radiation dose. As observed, mild hyperthermia (40–42 °C) causes an increase of the intratumoral blood flow, and subsequent tumor re-oxygenation facilitates the generation of reactive oxygen species by ionizing radiation, leading to increased DNA damage in the irradiated volume [37].

## 2. Results and Discussion

There are two methods for labeling metal oxide nanoparticles with radionuclides: surface adsorption on hydroxyl groups or incorporation of metallic radionuclide into the structure of the nanoparticle during its synthesis. Due to the higher stability, we decided to incorporate ^225^Ac into the nanoparticle structure. The ^225^Ac@Fe_3_O_4_ nanoparticles were successfully prepared by the co-precipitation method. Images from TEM studies (Figure 1) revealed that synthesized nanoparticles were generally spherical with an average 10.3 ± 1.2 nm diameter (*n* = 10 samples) and apparent aggregation in the vacuum environment of TEM. Due to the ultra-trace level of ^225^Ac concentration, it was not possible to investigate its incorporation into the crystal lattice. However, based on the similarity between Ac^3+^ and lanthanides cations, we can expect that Ac^3+^ incorporates into Fe_3_O_4_ structure in an analogous way as it was observed previously in the synthesis of SPIONs doped with Ho^3+^, Gd^3+^ and Eu^3+^ cations [38,39]. On the basis of X-ray studies, authors found that up to 2.5% of Ho^3+^ content, a new phase was formed in which some of the Fe^3+^ atoms were replaced by Ho^3+^. A schematic reaction taking place during the synthesis of doped SPIONs with ^225^Ac^3+^ should be as follows:
Fe^2+^ + *x*Fe^3+^ + (2 − *x*)^225^Ac^3+^ + 8OH^−^ → ^225^Ac_(2−*x*)_Fe_*x*_FeO_4_ + 4H_2_O(1)

Based on the results from the co-precipitation synthesis, the efficiency of ^225^Ac^3+^ (0.1–0.5 MBq) incorporation into 0.5 mg of Fe_3_O_4_ nanoparticles was very high with yield reaching 99.3 ± 0.7% (*n* = 10).

The stability of synthesized ^225^Ac@Fe_3_O_4_ nanoparticles was examined in physiological saline (0.9% NaCl), 1 mM PBS pH 7.4 and human serum for 10 days (Figure 2). The amount of released from nanoparticles mother radionuclide ^225^Ac and its decay products ^221^Fr and ^213^Bi was determined on a γ-spectrometer. The synthesized nanoparticles retained more than 98% of ^225^Ac and more than 90% of ^221^Fr and ^213^Bi over 10 days in 0.9% NaCl and 1 mM PBS. In human serum, the fraction of ^225^Ac released from ^225^Ac@Fe_3_O_4_ was still less than 2%, but the retention of ^221^Fr and ^213^Bi decreased to 70% after 10 days. Our results are comparable or slightly better, especially in the case of released daughter radionuclides, as observed previously on ^225^Ac-labeled titanium oxide [40] nanoparticles and Fe_3_O_4_ NPs labeled with ^223^Ra [28]. The obtained results can be interpreted on the basis of recoil energy in the ^225^Ac → ^221^Fr decay reaction and subsequent α decays [41]. The liberation of the recoiled radionuclides from ^225^Ac@Fe_3_O_4_ allows them to freely migrate in the body, causing toxicity to healthy tissues and decreasing the therapeutic dose delivered to the tumor. The renal toxicity induced by longer-lived decay product ^213^Bi is considered to be the major constraint to apply ^225^Ac in tumor therapy [42]. Since the recoil energy of ^225^Ac α-decay products varies from 105 to 135 keV, multilayers of lanthanide phosphate and gold would be needed to keep them within the nanoparticle [43]. The observed smaller release of daughter radionuclides ^221^Fr and ^213^Bi in PBS and saline solution suggests that Fe_3_O_4_ is an effective cation exchanger [44] that can reabsorb recoiled ^221^Fr^+^ and ^213^Bi^3+^ via hydroxyl groups on the NPs surface. This process can also explain the slightly lower stability of radiolabeled NPs in human serum, as in this medium hydroxyl groups on the surface of Fe_3_O_4_ NPs are blocked by the proteins from serum and this may prevent rebinding of released daughter radionuclides. It should be noted that these phenomena cannot be transferred from in vitro conditions to in vivo models where blood flow may rapidly dislocate the decay products from the surface of the ^225^Ac@Fe_3_O_4_ NPs, which might reduce the re-adsorption probability.

To direct ^225^Ac@Fe_3_O_4_ to cancer cells, we attached trastuzumab to the surface of nanoparticles. Trastuzumab is a monoclonal antibody that binds to HER2 receptors overexpressed in cancerous cells and it was used as a targeting vector of ^225^Ac@Fe_3_O_4_. A schematic procedure of nanoparticle surface functionalization with trastuzumab is presented in Figure 3. In the first step, 3-phosphonopropionic acid (CEPA) molecules were attached to the hydroxyl groups on the surface of Fe_3_O_4_ nanoparticles through the phosphonate groups to form strong covalent bonds. In the second step, carboxyl groups from ^225^Ac@Fe_3_O_4_-CEPA nanoparticles were activated to the NHS-ester, in the presence of EDC and sulfo-NHS, which is reactive towards primary amine groups from lysines available on the surface of trastuzumab, resulting in the formation of a very stable amide bond.

After each step of their surface functionalization with CEPA and trastuzumab, the synthesized nanoparticles, were analyzed by DLS to determine their hydrodynamic diameters and zeta (ζ) potentials (Table 1). As mentioned in the experimental part, for the physicochemical characterization of nanoparticles and their conjugates with CEPA and trastuzumab, the Fe_3_O_4_ nanoparticles were labeled with non-radioactive La^3+^ instead of ^225^Ac^3+^.

The La@Fe_3_O_4_ NPs size (91.4 ± 11.3 nm), determined by DLS, was notably larger than that measured by TEM (10.3 ± 1.2 nm). This difference was observed because TEM and DLS are different techniques since DLS determines the hydrodynamic diameter, which includes the solvation layers. In the case of TEM, dehydration of the nanoparticle surface takes place in the vacuum environment of TEM, thus the diameter of bare nanoparticles is measured. Determined polydispersity indexvalues for freshly synthesized La@Fe_3_O_4_ NPs, La@Fe_3_O_4_-CEPA and La@Fe_3_O_4_-CEPA-trastuzumab bioconjugate were below 0.3, which means that our probes can be considered as monodisperse. Moreover, for La@Fe_3_O_4_-CEPA, the PDI value is smaller than for the La@Fe_3_O_4_ NPs, which indicates that our linker stabilizes them and prevents the possible aggregation. The positive zeta potential of the synthesized La@Fe_3_O_4_-CEPA-trastuzumab (+17.2 mV) is related to the existence of protonated amine groups on the conjugated surface, while the La@Fe_3_O_4_-CEPA has negatively charged carboxyl groups from CEPA on the surface. A changing of zeta potential to positive clearly indicates attachment of the trastuzumab molecules to the surface.

Thermogravimetric analyses also provided experimental evidence for the presence of trastuzumab on the Fe_3_O_4_ NPs surface. The TGA thermograms of bare Fe_3_O_4_ and functionalized with CEPA or CEPA-trastuzumab are depicted in Figure 4. All thermograms indicate a small decrease of mass in the range of 25–100 °C that is associated with the desorption of physically adsorbed water. In the case of Fe_3_O_4_-CEPA NPs sample that was heated from 250 °C to 350 °C, a noticeable weight loss (~2.0%) is related to the release of 3-phosphonopropionic acid molecules. For nanoparticles with attached monoclonal antibodies (Fe_3_O_4_-CEPA-trastuzumab) the total loss of mass ~25% was observed after heating the sample up to 900 °C. Based on the obtained TGA results, the number of attached trastuzumab molecules per one nanoparticle was estimated. The calculation was performed under the assumption that each nanoparticle is spherical with an average diameter of 10 nm, as measured by TEM, and that the density of the material is 5.2 g·cm^−3^. The obtained results revealed that on average 8 trastuzumab molecules were coupled to the surface of one Fe_3_O_4_ nanoparticle.

In the radiometric method, functionalized and NHS-activated Fe_3_O_4_-CEPA NPs were reacted with ^131^I-trastuzumab, and coupling efficiency assessed by measuring the proportion of radioactivity bound to NPs in relation to initially added. The number of ^131^I-trastuzumab attached to each Fe_3_O_4_ nanoparticle was calculated by dividing the moles of ^131^I-trastuzumab bound to nanoparticles by the moles of used Fe_3_O_4_ NPs. By this method, it was estimated that 11 trastuzumab molecules were conjugated with one Fe_3_O_4_ nanoparticle, a value very similar to that determined by thermogravimetric analysis.

The use of SPIONs as ^225^Ac carriers gives us a unique possibility of multimodal treatment: α radionuclide therapy in combination with magnetic hyperthermia. In the case of superparamagnetic nanoparticles, the heat generation in an alternating magnetic field (AMF) is related to the coherent rotation of the magnetic moments for single domain NPs [45]. Thus, the heat generation may be due to the combined effect of the relaxation and hysteresis losses in the magnetic NPs. Due to the low content of ^225^Ac in ^225^Ac@Fe_3_O_4_, we also expected a hyperthermia effect in this material. As mentioned earlier, due to the inability of measurement of radioactive materials in hyperthermia equipment, the Fe_3_O_4_ nanoparticles were labeled with non-radioactive La^3+^ instead of ^225^Ac^3+^. The close-shell electron configuration of La^3+^ and Ac^3+^ results in almost identical chemical and magnetic properties for La^3+^ and Ac^3+^ cations. Figure 5 shows the changes in temperature depending on the magnetic field with varying frequency and amplitude for La@Fe_3_O_4_, La@Fe_3_O_4_-CEPA and La@Fe_3_O_4_-CEPA-trastuzumab. As can be seen in Figure 5, the application of high frequency 386.4 kHz, 488.3 kHz, and 632.6 kHz and high amplitude magnetic fields caused a spontaneous rise of the suspension’s temperature. The efficiency of the SPIONs in heat generation while exposed to an AMF is gauged via a specific absorption rate (SAR). The SAR values were estimated from the initial linear range of temperature as a function of time as presented in Figure 6. The modification of the La@Fe_3_O_4_ surface with small-molecule CEPA increases nearly twice the values of SAR. This is due to the greater dispersion of La@Fe_3_O_4_-CEPA, which increases the possibility of rotation of separate NPs as compared with their aggregates in the absence of CEPA stabilization. Further modification by attachment of large trastuzumab molecules significantly reduces the possibility of rotation and therefore reduces the Brownian relaxation. Despite this, the determined SAR value of 105 W·g^−1^ for La@Fe_3_O_4_-CEPA-trastuzumab bioconjugate should allow its application to clinical magnetic hyperthermia.

The cytotoxic effect of free ^225^Ac radionuclide, functionalized ^225^Ac@Fe_3_O_4_-CEPA nanoparticles, and ^225^Ac@Fe_3_O_4_-CEPA-trastuzumab radiobioconjugate was investigated on SKOV-3 cells, overexpressing the HER2-receptor, by their incubation with different activities of radiocompounds for 48 h and assessment of metabolic activity with the colorimetric MTS assay (Figure 7). SKOV-3 cell viability decreased with increasing radioactive doses of all compounds. The incubation with plain ^225^Ac induced some degree of cytotoxicity, although the surviving fraction at 50 kBq/mL was still 67.7 ± 4.3%, whereas for ^225^Ac@Fe_3_O_4_-CEPA nanoparticles and ^225^Ac@Fe_3_O_4_-CEPA-trastuzumab radiobioconjugate at the same dose it was 16.9 ± 6.6% and 7.9 ± 1.5%, respectively. Surprisingly, a higher than expected toxic effect of ^225^Ac@Fe_3_O_4_-CEPA nanoparticles was observed, especially in comparison to plain ^225^Ac. This was probably related to the fact that ^225^Ac@Fe_3_O_4_-CEPA and ^225^Ac@Fe_3_O_4_-CEPA-trastuzumab, in contrast to ^225^Ac^3+^ cations, sediment and adsorb on the cell surface in a long-term incubation process. The other explanation is the ability of small nanoparticles to impair the cell membrane integrity, which results in their internalization and accumulation in cytosolic vacuoles, as was already observed for PEGylated AuNPs [46]. This mechanism, although not specific, can increase the amount of delivered α-emitting ^225^Ac@Fe_3_O_4_-CEPA nanoparticles into cells, compared to ^225^Ac alone. Nonetheless, the determined *IC*_50_ values were 66.3 ± 1.1 kBq/mL, 7.6 ± 1.3 kBq/mL and 2.0 ± 1.1 kBq/mL for plain ^225^Ac, ^225^Ac@Fe_3_O_4_-CEPA and ^225^Ac@Fe_3_O_4_-CEPA-trastuzumab, respectively. These results indicate that ^225^Ac@Fe_3_O_4_-CEPA-trastuzumab is more toxic towards SKOV-3 cells than alone ^225^Ac@Fe_3_O_4_-CEPA nanoparticles without attached targeting vector. The assessed *IC*_50_ value for ^225^Ac@Fe_3_O_4_-CEPA-trastuzumab is higher in relation to ^225^Ac-trastuzumab on BT-474, MDA-MB-361 and modified SKOV-3.NMP2 cells [47,48], although it is difficult to directly compare the results obtained on cells with various HER2-receptor expression levels. Moreover, the cytotoxicity experiments were performed using different protocols, since in some studies the cells were incubated with radioactivity for up to 120 h, which is three times longer than in our case, and this could also have an impact on the determined IC_50_ values. On the other hand, ^225^Ac@Fe_3_O_4_-CEPA-trastuzumab seems to be around five times more effective in killing the SKOV-3 cells when compared to the ^225^Ac-DOTA-2Rs15d nanobody construct, earlier studied by our group [49]. It is also worth noting that only a small toxic effect was observed on SKOV-3 cells for non-radioactive Ho@Fe_3_O_4_-CEPA-trastuzumab nanoparticles, prepared by us and doped with non-radioactive Ho^3+^cations, at the applied concentration of 0.5–400 µg/mL, which is even higher than previously used in our studies [50].

The ex vivo biodistribution data of ^177^Lu@Fe_3_O_4_-CEPA and ^177^Lu@Fe_3_O_4_-CEPA-trastuzumab studied in SKOV-3 tumor-bearing SCID mice are presented in Figure 8A,B, respectively. As shown in both cases, the liver exhibited the highest accumulation of the injected radiotracers at all examined time points (24 and 72 h), followed by the spleen. Specifically, the liver and spleen uptake of the ^177^Lu@Fe_3_O_4_-CEPA was 45.49 ± 14.34% IA/g and 34.56 ± 9.23% IA/g, respectively, at 24 h p.i. and 38.43 ± 6.56% IA/g and 26.21 ± 9.37% IA/g, respectively, at 72 h p.i. In addition, the uptake of the ^177^Lu@Fe_3_O_4_-CEPA-trastuzumab in the liver was 45.82 ± 19.24% IA/g and 43.33 ± 8.80% IA/g at 24 h and 72 h p.i, respectively. The same applies to the spleen, with 42.81 ± 20.16% IA/g and 28.88 ± 8.47% IA/g at 24 h and 72 h p.i, respectively. The high accumulation of both compounds in the organs of the reticuloendothelial system (RES) (i.e., liver and spleen) observed in our study can be attributed to their hydrodynamic size, which was found to be 126.7 ± 12.0 nm for ^177^Lu@Fe_3_O_4_-CEPA and 216.3 ± 17.3 nm for ^177^Lu@Fe_3_O_4_-CEPA-trastuzumab via DLS measurement. It is well known that plasma proteins are adsorbed onto the surface of the intravenously injected radiolabeled nanoparticles and this adsorption is strongly related to their size. It has been shown [51] that nanoparticles with a hydrodynamic size smaller than 100 nm undergo 6% protein adsorption, whereas nanoparticles with a hydrodynamic size larger than 170 nm hydrodynamic size exhibit 23–34% protein adsorption onto their surface. Accordingly, since both liver and spleen are phagocytic-rich organs, they rapidly recognize and engulf opsonized radiolabeled nanoparticles. This is why radiotracers with a hydrodynamic size smaller than 10 nm undergo renal clearance, whereas the bigger sized nanoparticles with a hydrodynamic size higher than 200 nm accumulate mostly in the RES organs, resulting in fast and pronounced blood clearance from the blood circulation. Indeed, rapid blood clearance (<0.3% IA/g) and lower uptake in all other organs were observed (<2% IA/g for the ^177^Lu@Fe_3_O_4_-CEPA and <3% IA/g for ^177^Lu@Fe_3_O_4_-CEPA-trastuzumab) at both time points (24 and 72 h p.i.) studied here.

Referring to the tumor uptake, we see that the respective values of the % IA/g of both ^177^Lu@Fe_3_O_4_-CEPA and ^177^Lu@Fe_3_O_4_-CEPA-trastuzumab at 24 h and 72 h p.i. are quite low but show differences. Specifically, the tumor uptake of the ^177^Lu@Fe_3_O_4_-CEPA was 0.16 ± 0.07% IA/g at 24 h p.i. and remained practically stable up to 72 h p.i. (0.10 ± 0.05% IA/g). In this case, NPs accumulation in the tumor was non-specific, and may be attributed to the EPR effect. On the contrary, the uptake of the ^177^Lu@Fe_3_O_4_-CEPA-trastuzumab radiobioconjugates in the tumor was increased compared to the ^177^Lu@Fe_3_O_4_-CEPA (namely 0.44 ± 0.01% IA/g at 24 h p.i. and 0.39 ± 0.1% IA/g at 72 h p.i.), albeit quite low. Thus, despite the presence of the antibody used as a targeted moiety in the ^177^Lu@Fe_3_O_4_-CEPA-trastuzumab radiobioconjugates for the active targeting of the SKOV-3 tumor, no significant accumulation in the site of the tumor was observed. In accordance to our results, several papers have been reported in the literature concerning the low nanoparticle accumulation in tumors, despite the attachment of a targeting antibody [52,53,54,55]. For example, Chattopadhyay et al. [55] showed that intravenously injected ^111^In-labeled nanoparticles functionalized with trastuzumab (^111^In-Au-T) in MDA-MB-361 tumor-bearing CD-1 mice were rapidly taken up by the liver and spleen within 2 h p.i. and thus exhibited decreased tumor accumulation which reached the 1.23 ± 0.20% IA/g at 48 h p.i. compared to the non-targeted (NT) ones (2.20 ± 0.23% IA/g at 48 h p.i.). The ^111^In-Au-T nanoparticles showed much lower circulating ability on intravenous administration compared to the ^111^In-Au-NT ones, which took advantage of the EPR effect due to their longer residence time in the bloodstream. In the same context, Bartlett et al. [52] saw a negligible impact of the targeting moiety (namely transferrin) on the biodistribution of the ^64^Cu-labeled PEGylated nanoparticles, since both targeted and non-targeted nanoparticles exhibited identical tumor uptake (~1% IA/g) at 1 day p.i.

The low tumor uptake may be attributed to the increased hydrodynamic diameter of the intravenously injected ^177^Lu@Fe_3_O_4_-CEPA and ^177^Lu@Fe_3_O_4_-CEPA-trastuzumab radiotracers after their surface modification, resulting in their fast elimination from the blood circulation by the RES organs [56,57]. Kanazaki et al. [57] evaluated the tumor uptake of intravenously injected ^111^In-labeled Fe_3_O_4_ nanoparticles conjugated with anti-HER2 scFv of different sizes (i.e., 20, 50 and 100 nm) in N87 tumor-bearing mice at 24 h p.i. and saw that the tumor uptake was higher for the 20 nm-sized nanoparticles, namely 5.6 ± 0.1% IA/g for the 20 nm, 2.9 ± 0.1% IA/g for the 50 nm and 3.3 ± 0.2% IA/g for the 100 nm. At 24 h p.i. the uptake of the 20, 50 and 100 nm-sized radiolabeled nanoparticles in the liver was 10.6 ± 1.3% IA/g, 11.5 ± 0.5% IA/g, and 24.9 ± 4.5% IA/g, respectively, indicating an upward trend with increasing nanoparticle size. Our NPs increased hydrodynamic size (126.7 ± 12.0 nm and 216.3 ± 17.3 nm for ^177^Lu@Fe_3_O_4_-CEPA and ^177^Lu@Fe_3_O_4_-CEPA-trastuzumab, respectively) compared to the nanoparticles studied by Kazanaki et al., hence indicated higher liver uptake (45.49 ± 14.34% IA/g for ^177^Lu@Fe_3_O_4_-CEPA and 45.82 ± 19.24% IA/g for ^177^Lu@Fe_3_O_4_-CEPA-trastuzumab) and even lower tumor accumulation (0.16 ± 0.07% ^177^Lu@Fe_3_O_4_-CEPA and 0.44 ± 0.01% IA/g for ^177^Lu@Fe_3_O_4_-CEPA-trastuzumab) at 24 h p.i.

The density of the attached targeting moieties should also be considered as a potential factor that attributed to the low tumor uptake [54,58]. Colombo et al. [58] used trastuzumab-functionalized gold nanoparticles for active targeting of HER2-positive breast cancer and found that the best tumor accumulation and therapeutic efficacy was achieved with one antibody attached per nanoparticle, in contradiction to the general opinion that targeting efficiency is increased with increasing the number of antibodies per nanoparticle. The group also showed that in the case of low ligand densities (namely one antibody per nanoparticle) the contribution of active targeting was higher compared to passive targeting, while at higher ligand densities (namely two antibodies per nanoparticle) passive targeting might be dominant. The aforementioned trends are also reflected in the work of other groups [54,59]. The above conclusions are in accordance with our results with regard to low tumor uptake of ^177^Lu@Fe_3_O_4_-CEPA-trastuzumab after its intravenous administration, which may be attributed to the high number of antibodies (*ca*. 8-11) per NP in our experiments.

In a separate group of SKOV-3 tumor-bearing mice, the biodistribution of ^177^Lu@Fe_3_O_4_-CEPA-trastuzumab was assessed at 72 h post-intratumoral injection. The results of this experiment are presented as % IA/organ, due to the extremely small weight of the tumor (Figure 9). Tumor uptake was 6.27 ± 2.13%, with a tumor:blood ratio of ~40 and high Tumor-to-RES organ ratios (tumor:liver ~21, tumor:spleen ~209 and tumor:lung ~202). As opposed to the intravenous administration of the radiotracer, the intratumorally-injected NPs resulted in high tumor uptake, where practically all of the radiolabeled conjugate remained at the injection site, with all other organs demonstrating an uptake which is less than 1% of the injected activity. It should also be noted that bone uptake refers to the uptake of the whole skeleton, which represents 10% of the total body weight of the mouse.

## 3. Materials and Methods

### 3.1. General

The following reagents were purchased from Sigma-Aldrich (St. Louis, MO. USA), unless specified, and used directly without further purification: iron(II) chloride tetrahydrate (puriss. p.a., ≥99%), iron(III) chloride hexahydrate (ACS reagent, 97%), lanthanum(III) nitrate hexahydrate (99.9%), 3-phosphonopropionic acid (CEPA, 94%), *N*-(3-dimethylaminopropyl)-*N*′-ethylcarbodiimide hydrochloride (EDC, ≥99%), *N*-hydroxysulfosuccinimide sodium salt (sulfo-NHS, ≥98%), human serum (stored at −20 °C), 2-(*N*-morpholino)ethanesulfonic acid (MES, ≥99%), Iodogen (1,3,4,6-tetrachloro-3α,6α-diphenyl-glycoluril, Thermo Scientific, Rockford, IL), ammonia solution (25% NH_4_OH, CHEMPUR, Piekary Śląskie, Poland), sodium chloride (POCH, Gliwice, Poland), phosphate-buffered saline (PBS, Amresco, Solon, OH, USA). Trastuzumab was isolated from the commercial drug product (Herceptin^®^, Roche Pharmaceuticals, Basel, Switzerland) on Vivaspin 500 (Sartorius, Stonehouse, UK) concentrators with a 50 kDa cutoff membrane. Aqueous solutions were prepared using ultrapure deionized water (resistivity, 18.2 MΩ·cm) from Milli-Q filtering system (Merck, Darmstadt, Germany).

Actinium-225 was obtained by radiochemical separation from a ^229^Th source as described elsewhere [60,61]. The activity of ^225^Ac was quantified using high-resolution γ-spectrometry while it was in secular equilibrium with its daughters, typically the next day after sample collection. No-carrier-added sodium [^131^I]iodide with a specific activity of approximately 550 GBq mg^−1^, and [^177^Lu]lutetium(III) chloride with a specific activity of >370 GBq mg^−1^ were supplied from the POLATOM Radioisotope Centre (Świerk, Poland).

SKOV-3 (HER2-overexpressing) cells were purchased from the American Type Culture Collection (ATCC, Rockville, MD, USA) and cultured in McCoy’s 5A medium supplemented with 10% heat-inactivated fetal bovine serum, l-glutamine, streptomycin (100 μg/mL), and penicillin (100 IU/mL). All reagents for cell growth were purchased from Biological Industries (Beth Haemek, Israel). Cells were grown in a humidified atmosphere with 5% CO_2_ at 37 °C. Prior to their in vitro and in vivo use, cells were detached using trypsin-EDTA.

### 3.2. Techniques Used for Characterization of Synthesized Nanoparticles

The size, shape, and morphology of synthesized Fe_3_O_4_ nanoparticles and nanoparticle conjugates were characterized by transmission electron microscopy (Zeiss Libra 120 Plus, Stuttgart, Germany). The hydrodynamic diameter and zeta potential (ζ) of synthesized nanoparticles, before and after their modification with 3-phosphonopropionic acid (CEPA) and trastuzumab, were determined by dynamic light scattering (Zetasizer Nano ZS DLS, Malvern, UK). The modification of nanoparticle surface was also investigated by thermogravimetric analysis using SDT-Q600 Simultaneous TGA instrument (TA Instruments, New Castle, DE).

The concentration of purified trastuzumab was determined by UV-Vis spectrophotometer Evolution 600 (ThermoFisher Scientific, Madison, WI, USA) by using the molar extinction coefficient (ε) of 225000 M^−1^·cm^−1^ at 280 nm [62]. The radioactive samples were measured using γ-spectrometry on Coaxial High Purity Germanium (HPGe) detector (GX 1080) with multichannel analyzer DSA-1000 and Genie 2000 software (entire system from Canberra, Meriden, CT, USA). Magnetic hyperthermia was measured with nanoScale Biomagnetics D5 Series equipment with CAL1 CoilSet (Zaragoza, Spain). The Specific Absorption Rate (SAR) values were estimated using MaNIaC Controller software. The measurements were performed by altering the magnetic field in the frequency range of 386–633 kHz and with the magnetic flux density from 100 to 300 G. Samples were measured for 5 min or until the temperature reached 45 °C.

### 3.3. Synthesis of SPIONs Doped with ^225^Ac (^225^Ac@Fe_3_O_4_)

Superparamagnetic Iron Oxide Nanoparticles were synthesized following the protocol described earlier with minor modifications [63]. Briefly, equal volumes (250 µL) of 0.01 M FeCl_2_ × 4H_2_O and 0.02 M FeCl_3_ × 6H_2_O were mixed in an eppendorf tube, heated in a block-heater up to 80 °C for 5 min followed by 80 µL of 25% ammonia solution addition. The obtained mixture was stirred and heated for another 15 min during which a black precipitate of nanoparticles was formed. Synthesized magnetite nanoparticles were separated from the reaction mixture by a strong magnet, washed several times with deionized water and used for further studies on their surface modification.

The ^225^Ac doped SPIONs (^225^Ac@Fe_3_O_4_) were synthesized in a similar way as the non-radioactive ones by the co-precipitation and embedding of added ^225^Ac^3+^ cations in the magnetite crystal structure. The synthesis was performed in an eppendorf plastic tube (2 mL) by mixing 250 µL 0.01 M FeCl_2_ × 4H_2_O with 250 µL 0.02 M FeCl_3_ × 6H_2_O and 0.2–0.5 MBq of ^225^Ac as a chloride salt, followed by dropwise addition of 80 µL 25% ammonia until the pH of obtained solution reached 10. The formed dark solution was stirred for an additional 15 min, nanoparticles were separated by a strong magnet and washed several times with deionized water. The percentage of incorporated ^225^Ac was calculated from the radioactivity retained in the nanoparticles in comparison to the ^225^Ac initially added into the solution. The nanoparticles with incorporated ^177^Lu^3+^ (0.1–10 MBq) radionuclide were synthesized in a similar way and after surface modification used for mice biodistribution studies.

### 3.4. Stability Studies of ^225^Ac@Fe_3_O_4_ Nanoparticles

Synthesized nanoparticles were tested for the determination of leaching of the mother radionuclide ^225^Ac and its decay daughters as described earlier [40]. Briefly, a portion of radiolabeled nanoparticles was placed in a dialysis tube (D-Tube Dializer Midi with 3.5 kDa cut-off membrane, Novagen) and was dialyzed against 20 mL of 0.9% NaCl or 1 mM PBS for up to 10 days at room temperature. Every day a 1 mL aliquot was taken, measured on a γ-spectrometer and the percentage of liberated activity from each radionuclide was determined by its characteristic gamma-peaks. Stability in human serum (HS) was determined by preparing 10 identical samples in which 50 μL of nanoparticles were suspended in 1 mL of HS, vortexed and incubated at 37 °C for 10 days. Each day, a probe was centrifuged and 500 μL of HS measured for ^225^Ac and its decay daughters.

### 3.5. Synthesis of ^225^Ac@Fe_3_O_4_-CEPA-Trastuzumab Bioconjugate

The attachment of trastuzumab biomolecules to the surface of previously synthesized non-radioactive magnetite nanoparticles or ^225^Ac@Fe_3_O_4_ nanoparticles was a two-step procedure. In a first step, 5 mL of 0.1 M NaOH was mixed with 5 mL of 25 mg/mL of 3-phosphonopropionic acid (CEPA), comprising the phosphonic and carboxyl groups at each end (PO(OH)_2_-CH_2_-CH_2_-COOH), followed by dropwise addition of 1 mL 2.5 mg/mL solution of SPIONs or ^225^Ac@Fe_3_O_4_. The suspension was dispersed for 20 min in an ultrasonic bath and the nanoparticles were separated from unreacted 3-phosphonopropionic acid by a strong magnet and washed a few times with deionized water. The thus formed surface-modified nanoparticles were dispersed in 1.5 mL of water and mixed with 0.5 mL of 0.5 M MES buffer pH 6.0, containing 12.1 mg (105.1 μmol) of sulfo-NHS and 4 mg (20.9 μmol) of EDC. The solution was stirred for 30 min, during which an active NHS-ester was formed on the carboxylic groups of the previously attached 3-phosphonopropionic acid. Next, the nanoparticles were separated from the reaction mixture, washed with deionized water and resuspended in 1 mL of 1 mM PBS pH 8.0 followed by dropwise addition of trastuzumab solution (250 μg, 1.7 nmoles in 1 mL PBS) and reacted overnight at 4 °C. After this time the product was separated from the solution using a solid magnet, washed with deionized water and used for further studies.

### 3.6. Synthesis of La@Fe_3_O_4_-CEPA Nanoparticles and La@Fe_3_O_4_-CEPA-Trastuzumab Bioconjugate for TEM, DLS and Hyperthermia Studies

To evaluate the effect of heating efficiency, magnetic hyperthermia measurements were performed using a suspension of La@Fe_3_O_4_-CEPA nanoparticles without and with bioconjugation. Due to safety reasons, the Fe_3_O_4_ nanoparticles were labeled with non-radioactive La^3+^ instead of ^225^Ac^3+^. The close-shell electron configuration of La^3+^ and Ac^3+^ results in the almost identical chemical and magnetic properties of La^3+^ and Ac^3+^ cations. When compared to the synthesis of ^225^Ac@Fe_3_O_4_ NPs, the only difference in the synthesis of La@Fe_3_O_4_ NPs was the addition of 1% La^3+^ in the form of La(NO_3_)_3_ to the reaction mixture instead of no-carrier-added ^225^Ac. The CEPA and trastuzumab modification was carried out identically as described above for the ^225^Ac@Fe_3_O_4_-CEPA-trastuzumab nanobioconjugate.

### 3.7. Estimation of the Number of Attached Trastuzumab Per One Magnetite Nanoparticle

The average number of trastuzumab molecules attached to each SPION was determined by thermogravimetric analysis in which 5 mg of dried powder nanoparticles were placed in a TGA furnace and heated. Changes in the mass of synthesized nanoparticle samples were measured in the temperature range from 25 °C to 900 °C under the flow of inert gas (nitrogen) through the furnace at the rate of 100 mL/min. Analyses of mass loss as a function of rising temperature (10 °C/min) were prepared using TA Universal Analysis software.

The second method relied on coupling ^131^I-labeled trastuzumab onto nanoparticles by using the procedure described earlier with some modifications [64]. Briefly, 2 mg of trastuzumab in 200 µL of 0.05 M PBS was labeled with ^131^I (30–40 MBq) by using tubes coated with 10 µg of dried Iodogen. After incubation for 10 min at room temperature, the radioiodinated ^131^I-trastuzumab was purified on PD-10 columns filled with Sephadex G-25 resin (GE Healthcare, Life Sciences). In the next step, 250 μg of ^131^I-trastuzumab was added to SPION nanoparticles coated with 3-phosphonopropionic in which carboxylic groups were transformed into reactive NHS-ester as described above, and the entire mixture was stirred overnight. The next day, coated nanoparticles were separated from the solution with a strong magnet, washed several times with deionized water and dispersed in PBS. The radioactivity of each fraction was measured by γ-spectrometry and the binding efficiency of ^131^I-trastuzumab to SPIONs was assessed from the proportion of radioactivity coupled to nanoparticles. The number of ^131^I-trastuzumab attached to each Fe_3_O_4_ nanoparticle was calculated by dividing the moles of ^131^I-trastuzumab bound to nanoparticles by the moles of used Fe_3_O_4_ NPs.

### 3.8. Magnetic Hyperthermia Studies

Magnetic hyperthermia was measured with nanoScale Biomagnetics D5 Series equipment with CAL1 CoilSet. The SAR values were determined using the MaNIaC controller software. Briefly, 5 mg of bare Fe_3_O_4_ nanoparticles doped with 1% of La^3+^ or NPs with bound CEPA and trastuzumab were dispersed into 0.5 mL of water. The samples were inserted into the copper coil (thermostated at 18–20 °C) and the measurement was carried out. The temperature change in time was monitored and recorded for the frequency range of 386–633 kHz and magnetic flux density from 100 to 300 G. The small volume was chosen to reduce heating inhomogeneity. The time required to reach 45 °C was monitored in order to evaluate the heating rate and SAR values of the suspension.

### 3.9. In Vitro Cytotoxicity Assay

The cytotoxicity of the synthesized ^225^Ac@Fe_3_O_4_-CEPA-trastuzumab radiobioconjugate, ^225^Ac@Fe_3_O_4_-CEPA NPs and ^225^Ac radionuclide were studied on the SKOV-3 cell line. Cells were seeded in 96-well plates at a density of 3 × 10^3^ cells per well. The next day, the cells were washed twice with cold PBS followed by addition in triplicates of 100 μL of radiocompounds (0.4–100 kBq/mL) suspended in cell culture medium and incubated for 48 h at 37 °C. To assess cell metabolic activity, the CellTiter-96^®^ AQueous-Non-Radioactive (Promega, Mannheim, Germany) MTS assay was used. The absorbance in wells was measured at 490 nm using a micro-plate-reader (Apollo 11LB913, Berthold, Bad Wildbad, Germany). The cell viability was expressed in percent by normalization to cells grown in medium only. The concentration of radioactive species in kBq/mL required to inhibit the cell viability by 50% (IC_50_) was determined by analyzing viability curves with GraphPad Prism 5.1 (GraphPad Software, San Diego, CA, USA).

### 3.10. Ex Vivo Biodistribution

Animals used for the biodistribution studies were obtained from the breeding facilities of the Institute of Biosciences and Applications, NCSR “Demokritos”. The experimental animal facility is registered according to the Greek Presidential Decree 56/2013 (Reg. Number: EL 25 BIO 022), in accordance with the European Directive 2010/63 which is harmonized with national legislation, on the protection of animals used for scientific purposes. All applicable national guidelines for the care and use of animals were followed. Animal protocols were approved by the Ethical Committee for Animal Experiments of the Institute of Nuclear and Radiological Studies and Technology, Energy and Safety, of the N.C.S.R. ‘Demokritos’. Due to the limited access of ^225^Ac and no permission of working with this radionuclide at the N.C.S.R. “Demokritos”, the biodistribution experiments were performed with nanoparticles doped with ^177^Lu radionuclide. Female SCID mice were inoculated subcutaneously in the right hind leg with 4 × 10^6^ SKOV-3 cells in 50% Matrigel (BD Biosciences, Franklin Lakes, NJ, USA). Groups of xenografted mice (*n* = 3 per group, 2 assessment time-points) were injected intravenously via the tail vein with 100.5 ± 5.2 kBq of ^177^Lu@Fe_3_O_4_-CEPA or ^177^Lu@Fe_3_O_4_-CEPA-trastuzumab radiobioconjugate suspended in 100 μL of 1 mM PBS. Additionally, the same radiobioconjugate was intratumorally injected in a separate group of three tumor-bearing mice. At 24 and 72 h post-intravenous injection and at 72 h post-intratumoral injection, the mice were euthanized, followed by isolation of major organs and tissues, which were weighed and counted for radioactivity along with injection standards in an automatic γ-counter (Cobra II, Canberra, Packard). Results are expressed as a percentage of injected activity per gram of tissue (%IA/g) for the first set of experiments, and percentage of injected activity per organ (%IA/organ) for the intratumorally-injected group of mice.

## 4. Conclusions

We have shown that Fe_3_O_4_ labeled with ^225^Ac and functionalized with trastuzumab can be used to deliver and retain ^225^Ac and its daughter products at a target site. Obtained ^225^Ac@Fe_3_O_4_-CEPA-trastuzumab has been shown to have high receptor affinity toward ovarian cancer cells expressing HER2 receptor in-vitro and, as demonstrated by in-vivo studies, exhibits properties suitable for the treatment of cancer cells by intratumoral or post-resection injection. The intravenous injection of the ^225^Ac@Fe_3_O_4_-CEPA-trastuzumab radiobioconjugate for ovarian or breast cancer treatment should be excluded due to its relatively large size and the resulting high accumulation of radiolabeled nanoconstruct in the liver, lungs and spleen, causing rapid removal from the blood, exposing these organs to a toxic α radiation and very low tumor accumulation. Presented results are the step of our research towards the synthesis of SPION-Tmab radiopharmaceuticals, through the incorporation of α-emitter in the iron oxide magnetic core. With such a system we expect to achieve both: active targeting and multimodal action by simultaneous internal and localized irradiation and magnetic hyperthermia of specific cancers. However, due to the micrometer range of α-particles, this form of therapy seems to be most suitable for destroying single micrometastatic cancer cells or small volume disseminated disease, rather than larger tumor masses.

## Figures and Tables

**Figure 1 molecules-25-01025-f001:**
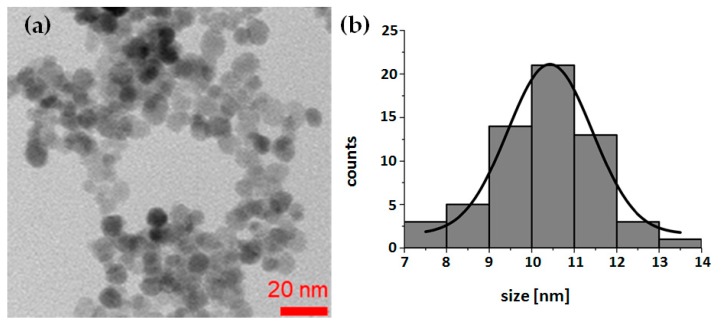
Representative transmission electron microscopy (TEM) image of synthesized Fe_3_O_4_ nanoparticles (**a**), and a histogram of nanoparticles sized from TEM image (**b**).

**Figure 2 molecules-25-01025-f002:**
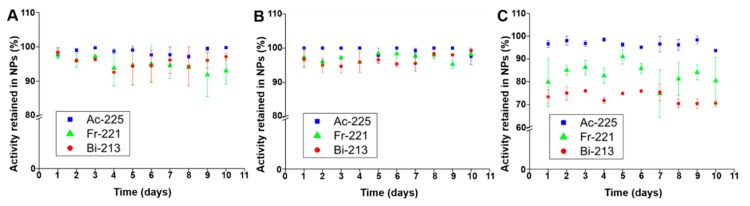
Stability studies of ^225^Ac@Fe_3_O_4_ nanoparticles in 0.9% NaCl (**A**), 1 mM phosphate-buffered saline (PBS) (**B**), and human serum (**C**).

**Figure 3 molecules-25-01025-f003:**
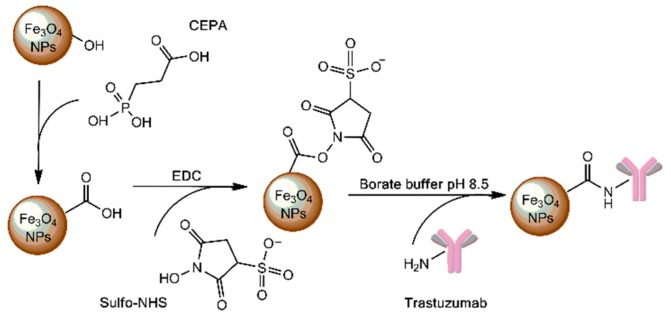
Scheme of the Fe_3_O_4_ nanoparticles surface functionalization with 3-phosphonopropionic acid (CEPA) and trastuzumab molecules.

**Figure 4 molecules-25-01025-f004:**
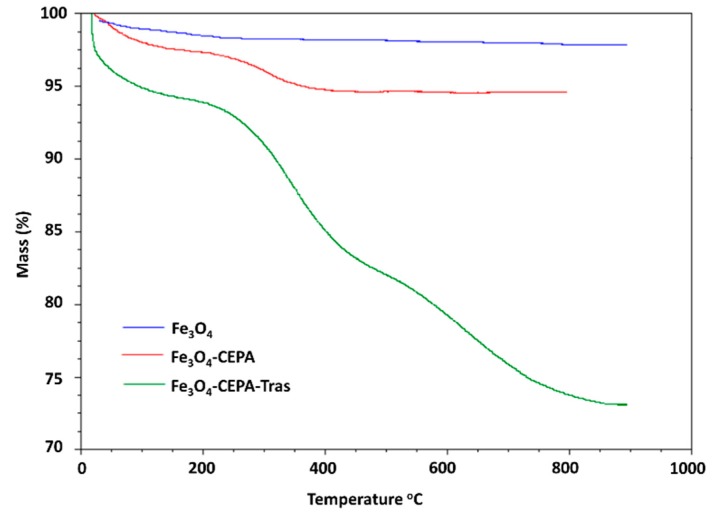
Representative thermograms of bare Fe_3_O_4_ nanoparticles and functionalized with CEPA and trastuzumab molecules.

**Figure 5 molecules-25-01025-f005:**
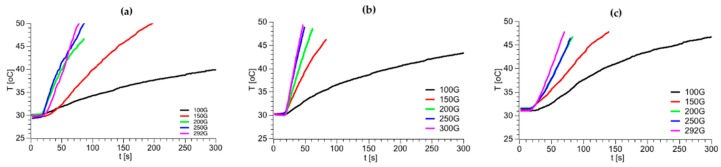
Heating of La@Fe_3_O_4_ (**a**), La@Fe_3_O_4_-CEPA (**b**) and La@Fe_3_O_4_-CEPA@-trastuzumab (**c**) at 632.6 kHz frequency of alternating magnetic field and various magnetic amplitude.

**Figure 6 molecules-25-01025-f006:**
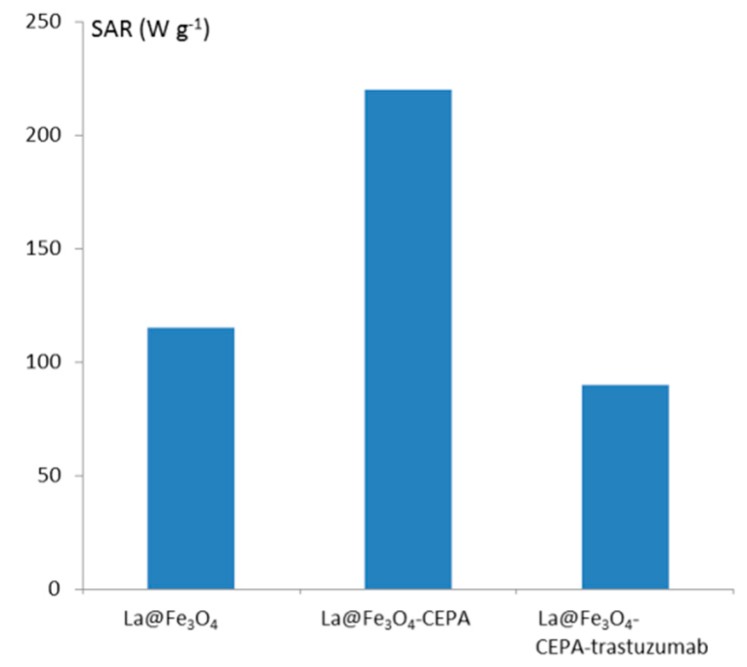
SAR values for unmodified and modified La@Fe_3_O_4_ nanoparticles.

**Figure 7 molecules-25-01025-f007:**
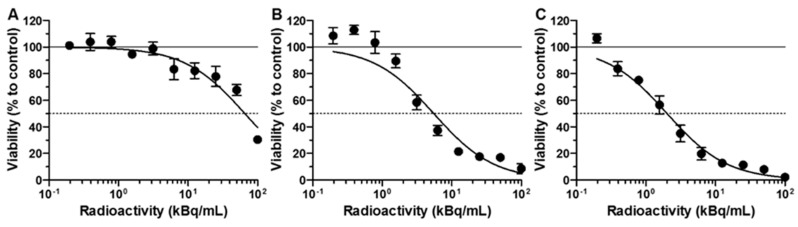
Viability of SKOV-3 cells after treatment with different radioactive doses of ^225^Ac (**A**), ^225^Ac@Fe_3_O_4_-CEPA (**B**), and ^225^Ac@Fe_3_O_4_-CEPA-trastuzumab (**C**). Cells were incubated for 48 h after which their viability was measured using MTS assay. Data are expressed as a percentage of control cells.

**Figure 8 molecules-25-01025-f008:**
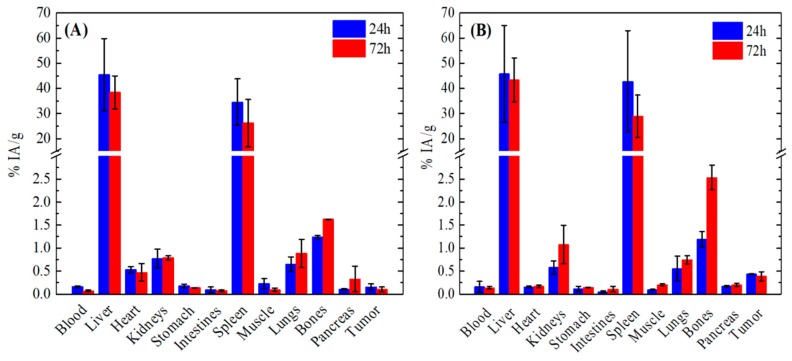
Ex vivo biodistribution data of the (**A**) ^177^Lu@Fe_3_O_4_-CEPA and (**B**) ^177^Lu@Fe_3_O_4_-CEPA-trastuzumab at 24 h and 72 h p.i. in SKOV-3 tumor-bearing SCID mice. The uptake in each organ is expressed as percentage injected activity per gram of tissue (% IA/g). For each time point, three mice were studied, thus the results are expressed as the mean ± SD.

**Figure 9 molecules-25-01025-f009:**
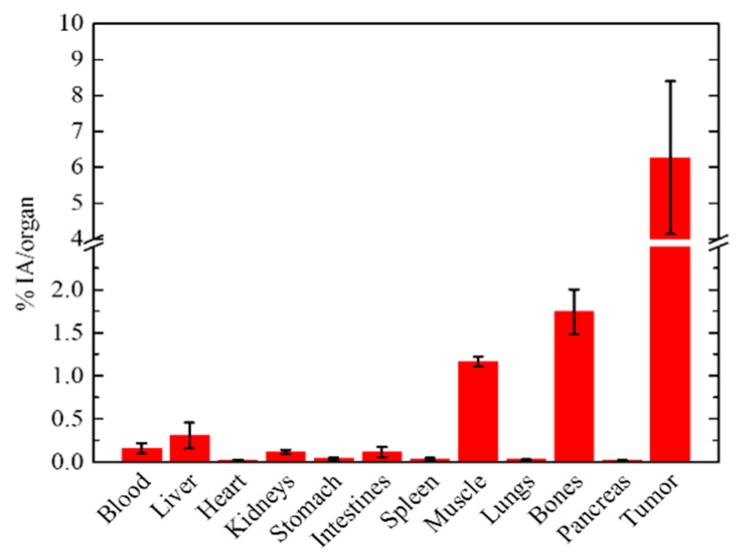
Ex vivo biodistribution data of ^177^Lu@Fe_3_O_4_-CEPA-trastuzumab at 72 h p.i. in SKOV-3 tumor-bearing SCID mice, after intratumoral injection.

**Table 1 molecules-25-01025-t001:** The hydrodynamic diameters, polydispersity index (PDI) and zeta (ζ) potentials of bare La@Fe_3_O_4_ NPs, La@Fe_3_O_4_-CEPA and La@Fe_3_O_4_-CEPA-trastuzumab determined by the Dynamic Light Scattering (DLS) method.

	La@Fe_3_O_4_	La@Fe_3_O_4_-CEPA	La@Fe_3_O_4_-CEPA-Trastuzumab
Hydrodynamic diameter (nm)	91.4 ± 11.3	126.7 ± 12.0	216.3 ± 17.3
Polydispersity index	0.237 ± 0.009	0.164 ± 0.017	0.268 ± 0.051
Zeta potential (mV)	+20.2 ± 0.8	−10.7 ± 0.2	+17.2 ± 0.3

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
