# Peer review of "Trastuzumab Conjugated Superparamagnetic Iron Oxide Nanoparticles Labeled with 225Ac as a Perspective Tool for Combined α-Radioimmunotherapy and Magnetic Hyperthermia of HER2-Positive Breast Cancer"

_molecules, 2020, doi:10.3390/molecules25051025_

Round 1
Reviewer 1 Report
The authors constructed magnetic NPs comprising α emitter 225Ac and trastuzumab (225Ac@Fe3O4-CEPA-trastuzumab). The bounding of the conjugate was investigated with TEM, FGA and 131I labelling. Based on the results, the authors concluded that the bioconjugate has a high cytotoxic effect toward SKOV-3 ovarian cancer cells expressing HER2 receptor in-vitro. The in-vivo studies indicated possible application for treatment of cancer by intratumoral or post-resection injection however the intravenous injection was excluded due to its high accumulation in the liver, lungs and spleen. The results showed the high value of SAR (ex vivo) may enable magnetic hyperthermia, in addition to radiotherapy.
In comparison to previous reports, the authors used alpha-emitter 225Ac instead of the β- emitter 177Lu utilizing its short radiation range and high RBE.
This is an interesting paper providing new knowledge.
I have no substantial negative comments.
The value or research could be increased with in vivo studies (e.g. heating effect), that I hope will follow.
Comments:
Could the authors compare the measured ex-vivo heating (5 mg Fe3O4 in 0.5 mL of water) with the expected in vivo tumor heating (considering blood volume, blood flow, accumulation of the contrast in tumor etc)? If the heating effect in vivo is small then the authors should tone down the last sentence: “a new very perspective combination of radionuclide therapy with magnetic hyperthermia.” Without results in vivo or at least an estimate, this is an overstatement.
Minor grammar mistakes, for example:
Due to the higher stability, we decided on the “second” method …
“In” this way, the uptake of bioconjugated…
“in” 247 632.6 kHz frequency etc. etc.
Author Response
Comments:
Could the authors compare the measured ex-vivo heating (5 mg Fe3O4 in 0.5 mL of water) with the expected in vivo tumor heating (considering blood volume, blood flow, accumulation of the contrast in tumor etc)? If the heating effect in vivo is small then the authors should tone down the last sentence: “a new very perspective combination of radionuclide therapy with magnetic hyperthermia.” Without results in vivo or at least an estimate, this is an overstatement.
In the case of hyperthermia, it is very difficult to translate in-vitro results into in-vivo tests. We need to take into account a number of parameters such as tissue type, tumor accumulation and internalization of magnetic nanoparticles in the cell, as well as blood flow that the reviewer mentioned. However, a SAR value of 105 W/g determined for our bioconjugate should allow to heat the tumor tissue up to 43-45oC as described by other authors for SPION conjugates having SAR around 100 W/g. In the continuation of our work, we are planning such studies on an animal model with an incubated tumor.
Minor grammar mistakes, for example: Due to the higher stability, we decided on the “second” method …; “In” this way, the uptake of bioconjugated…; “in” 247 632.6 kHz frequency etc. etc.
Thank you for highlighting these grammar mistakes which were corrected in the current version of the manuscript.
Reviewer 2 Report
Few comments and questions that authors may consider:
How does the 225Ac@Fe3O4 nanoparticles were characterized? What is the reproducibility of the manufacturing method? Does the change of the nanoparticle size during stability testing has been evaluated? Aggregation of NPs during storage has been tested? Abbreviations should be explained when first used How can be commented the intense hydration of the nanoparticle surface (line 191) and significant NP diameter variations? What PDI values were determined and how that could be commented? Characterization of nanoparticles with attached monoclonal antibodies (Fe3O4-CEPA-trastuzumab) – how many possible versions of attachment were determined? How the comparability study of trastuzumab molecules before vs. after attachment was implemented? What is the max theoretically possible number of trastuzumab molecules on the surface of the nanoparticle (line 210) – just for understanding of 8 molecules. The statement “…cytotoxicity of our compound is indeed HER2-mediated due to trastuzumab molecules attached to the surface of the nanoparticles.” (line 267) has to be supported by some reliable data or the modification of the wording is recommended. The statement on line 317 “…NPs accumulation in the tumor occurred in a nonspecific passive way, taking advantage of the enhanced permeability and retention effect.” lacks the backing by some experimental data, thus rewording is recommended. Conclusions are limited to repetition of some results and could be modified.Author Response
Reviewer 2
How does the 225Ac@Fe3O4 nanoparticles were characterized?
We do not have ability to directly characterize 225Ac@Fe3O4 nanoparticles as they are labeled with radioactive α-emitter 225Ac that may contaminate used by us highly specialized equipment. Therefore, all measurements and physicochemical characterization (TEM, DLS, TGA, etc.) were performed with the use of nanoparticles doped with non-radioactive lanthanum La@Fe3O4 NPs, those nanoparticles were synthesized as described in the manuscript at the same conditions as radioactive NPs.
What is the reproducibility of the manufacturing method?
Synthesis of the nanoparticles was performed many times and based on TEM analyses (n = 10) the reproducibility of the nanoparticles size and shape is confirmed. We added to the manuscript information that TEM analyses were performed for samples number of 10 at least.
Does the change of the nanoparticle size during stability testing has been evaluated?
When it comes to stability of 225Ac@Fe3O4 nanoparticles we were not able to measure the size of the radiolabeled nanoparticles as it is described above (point no. 1.).
Aggregation of NPs during storage has been tested?
Our studies with non-radioactive NPs doped with La revealed their stability and non-aggregation over time of 1-2 weeks when they are stored in water. However, our intention is to attach targeting biomolecule immediately after the synthesis of 225Ac-labeled nanoparticles.
Abbreviations should be explained when first used
done
How can be commented the intense hydration of the nanoparticle surface (line 191) and significant NP diameter variations?
The particles size determined by DLS was significantly larger than that observed by TEM. This is caused by fact that the DLS technique measures the mean hydrodynamic diameter of the nanoparticle core bounded by solvation layers, and this hydrodynamic diameter is affected by the viscosity and concentration of the medium. Strong hydration of 225Ac-Fe3O4 nanoparticles occurs due to the presence of hydroxyl groups on the surface that interact through hydrogen bonds formation with water molecules. TEM, however, gives the diameter of the core alone. As mentioned in the text in the case of TEM, dehydration of the nanoparticle surface takes place in the vacuum environment of TEM, thus the diameter of bare nanoparticles is measured.
What PDI values were determined and how that could be commented?
PDI and DLS measurements were performed on non-radioactive nanoparticles doped with La. All the measurements for DLS were performed for the concentration of compounds equal 0.01mg/mL. Determined by us PDI values for freshly synthesized La-Fe3O4, La- Fe3O4-CEPA and La- Fe3O4-CEPA trastuzumab bioconjugate were below 0.3, which means that size of our probes can be consider as monodisperse. Moreover, for La- Fe3O4-CEPA, PDI value is smaller than for the La- Fe3O4 NPs, which means that our linker stabilizes them well and prevents the possible aggregation.
As described previously, we do not plan to store our compounds, but we also determined the size with DLS method after 28 days for La- Fe3O4 and La- Fe3O4-CEPA. We have noticed a slightly bigger size values and increase of the PDI value. It is worth mentioning that PDI values are still around 0.3. In comparison to the measurements performed right after the synthesis, we assume that our NPs are slowly aggregating in time. We did not store bioconjugate with Trastuzumab, as after 28 days degradation of antibody may occur.
The appropriate sentence was added to the manuscript.
Characterization of nanoparticles with attached monoclonal antibodies (Fe3O4-CEPA-trastuzumab) – how many possible versions of attachment were determined? How the comparability study of trastuzumab molecules before vs. after attachment was implemented? What is the max theoretically possible number of trastuzumab molecules on the surface of the nanoparticle (line 210) – just for understanding of 8 molecules
The TGA analyses and radiometric assays for determination of number of attached trastuzumab molecules on the surface of NPs were performed several times that allow us to determine (according to described in the manuscript protocols) that average number of trastuzumab molecules by TGA analysis was 8 and by radiometric assay 11. The max theoretical possible number of trastuzumab molecules on the NPs surface is difficult to estimate as it depends not only on the size of NPs, but also on how many CEPA molecules are attached to the NPs surface and also how many of them will activate to the NHS-ester that is reactive towards primary amines from lysines accessible from trastuzumab.
Considering only simple geometric calculations, it can be assumed that a maximum of about 25 trastuzumab molecules can be attached to one 10 nm 225Ac-Fe3O4 nanoparticle (assuming a trastuzumab diameter of 2 nm).
The statement “…cytotoxicity of our compound is indeed HER2-mediated due to trastuzumab molecules attached to the surface of the nanoparticles.” (line 267) has to be supported by some reliable data or the modification of the wording is recommended.
Based on the obtained by us results and determined IC50 values it seems that 225Ac@Fe3O4-CEPA-trastuzumab was more toxic at in vitro conditions than alone 225Ac@Fe3O4-CEPA nanoparticles without attached targeting vector. We have changed sentence in the manuscript according to reviewers comment and current sentence is: “These results indicate that 225Ac@Fe3O4-CEPA-trastuzumab is more toxic towards SKOV-3 cells than alone 225Ac@Fe3O4-CEPA nanoparticles without attached targeting vector.”
The statement on line 317 “…NPs accumulation in the tumor occurred in a nonspecific passive way, taking advantage of the enhanced permeability and retention effect.” lacks the backing by some experimental data, thus rewording is recommended.
The authors have proceeded to rewrite the sentence in a more appropriate manner (lines 325 – 327).
Conclusions are limited to repetition of some results and could be modified.
The conclusions were changed as suggested by the reviewer.